# AN ALARM SYSTEM FOR SEGMENTATION ALGORITHM BASED ON SHAPE MODEL

## ABSTRACT

It is usually hard for a learning system to predict correctly on the rare events, and there is no exception for segmentation algorithms. Therefore, we hope to build an alarm system to set off alarms when the segmentation result is possibly unsatisfactory. One plausible solution is to project the segmentation results into a low dimensional feature space, and then learn classifiers/regressors in the feature space to predict the qualities of segmentation results. In this paper, we form the feature space using shape feature which is a strong prior information shared among different data, so it is capable to predict the qualities of segmentation results given different segmentation algorithms on different datasets. The shape feature of a segmentation result is captured using the value of loss function when the segmentation result is tested using a Variational Auto-Encoder(VAE). The VAE is trained using only the ground truth masks, therefore the bad segmentation results with bad shapes become the rare events for VAE and will result in large loss value. By utilizing this fact, the VAE is able to detect all kinds of shapes that are out of the distribution of normal shapes in ground truth (GT). Finally, we learn the representation in the one-dimensional feature space to predict the qualities of segmentation results. We evaluate our alarm system on several recent segmentation algorithms for the medical segmentation task. The segmentation algorithms perform differently on different datasets, but our system consistently provides reliable prediction on the qualities of segmentation results.

## 1 INTRODUCTION

A segmentation algorithm usually fails on the rare events, and it is hard to fully avoid such issue. The rare events may occur due to the limited number of training data. To handle it, the most intuitive way is to increase the number of training data. However, the labelled data is usually hard to collect, e.g., to fully annotate a 3D medical CT scan requires professional radiology knowledge and several hours of work. In addition, the human labelling is unable to cover all possible cases. Previously, various methods have been proposed to make better use of training data, like sampling strategies paying more attention to the rare events (Wang et al., 2018). But still it may fail on the rare events which never occur in the training data. Another direction is to increase the robustness of the segmentation algorithm to the rare events. Kendall & Gal (2017) proposed the Bayesian neural network which can model the uncertainty as an additional loss to make the algorithm more robust to noisy data. These kinds of methods make the algorithm insensitive to certain types of perturbations, but the algorithms may still fail on other perturbations.

Since it is hard to completely prevent the segmentation algorithm from failure, we consider to detect the failure instead: build up an alarm system cooperating with the segmentation algorithm, which will set off alarms when the system finds that the segmentation result is not good enough. This task is also called as quality assessment. Several works have been proposed in this field. Jungo et al. (2018) applied Bayesian neural network to capture the uncertainty of the segmentation result and set off alarm based on that uncertainty. However, this system also suffers from rare events since the segmentation algorithms often make mistakes confidently on some rare events (Xie et al., 2017). Kohlberger et al. (2012) provided an effective way by projecting the segmentation results into a feature space and learn from this low dimension space. They manually design several heuristic features, e.g., size, intensity, and assume such features would indicate the quality of the segmentation results. After projecting the segmentation results into the feature space, they learned a classifier to

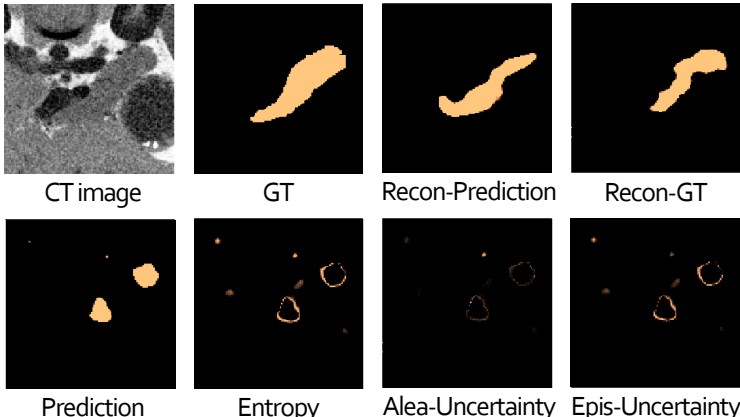

| CT image | GT | Recon-Prediction | Recon-GT |

| Prediction | Entropy | Alea-Uncertainty | Epis-Uncertainty |

Figure 1: The visualize on an NIH CT dataset for pancreas segmentation. The figures Recon-Prediction and Recon-GT are reconstruction results from prediction and GT by VAE network respectively. The Dice score between the GT and prediction is 47.06 while the Dice score between the prediction and Recon-Prediction is 47.25. In our method, we use the later Dice score to predict the former real Dice score which is usually unknown at inference phase in real applications. This case shows how these two Dice scores are related to each other. On the other hand, for uncertainty based methods, different kinds of uncertainty distribute mainly on the boundary of predicted mask, which makes it a vague information when detecting the failure cases.

predict its quality. Since the feature space is of relative low dimension now, it is able to distinguish good segmentation results from bad ones directly. In a reasonable feature space, when the segmentation algorithm fails, the failure output will be far from the ground truth. So the main problems is what these good features are and how to capture them. Many features that Kohlberger et al. (2012) selected are actually less related with the quality of segmentation results, e.g., size.

In our system, we choose a more representative feature, i.e., the shape feature. The shape feature is important because the segmenting objects (foreground in the volumetric mask) often have stable shapes among different cases, especially in 3D. So the shape feature is supposed to provide a strong prior information for judging the quality of a segmentation result, i.e., bad segmentation results tend to have bad shapes and vice versa. Furthermore, to model the prior from the segmentation mask space is much easier than in the image space and the shape prior can be shared among different datasets while the features like image intensity are affected by many factors. That means the shape feature can deal with not only rare events but also different data distributions in the image space, which shows great generalization power and potential in transfer learning. We propose to use the Variational Auto-Encoder(VAE) (Kingma & Welling, 2013) to capture the shape feature. The VAE is trained on the ground truth masks, and afterwards we define the value of the loss function as the shape feature of a segmentation result when it is tested with VAE network. Intuitively speaking, after the VAE is trained, the bad segmentation results with bad shapes are just rare events to VAE because it is trained using only the ground truth masks, which are under the distribution of normal shapes. Thus they will have larger loss value. In this sense we are utilizing the fact that the learning algorithms will perform badly on the rare events. Formally speaking, the loss function, known as the variational lower bound, is optimized to approximate the function $\log P(Y)$ during the training process. So after the training, the value of the loss function given a segmentation result $\hat{Y}$ is close to $\log P(\hat{Y})$, thus being a good definition for the shape feature.

In this paper, we proposed a VAE-based alarm system for segmentation algorithms. The qualities of the segmentation results can be well predicted using our system. To validate the effectiveness of our alarm system, we test it on multiple segmentation algorithms. These segmentation algorithms are trained on one dataset and tested on several other datasets to simulate when the rare events occur. The performance for the segmentation algorithms on the other datasets (rather than the training dataset) drops quickly but our system can still predict the qualities accurately. We compare our system with other alarm systems on the above tasks and our system outperforms them by a large margin, which

shows the importance of shape feature in alarm system and the great power of VAE in capturing the shape feature.

## 2 RELATED WORK

Kendall & Gal (2017) employed Bayesian neural network (BNN) to model the aleatoric and epistemic uncertainty. Afterwards, Kwon et al. (2018) applied the BNN to calculate the aleatoric and epistemic uncertainty on medical segmentation tasks. Jungo et al. (2018) utilized the BNN and model another kind of uncertainty based on the entropy of segmentation results. They calculated a doubt score by summing over weighted pixel-vise uncertainty. However we can see from Figure 1 that when the segmentation algorithm fails to provide correct prediction, the uncertainty still distributes mainly on the boundary of the wrong segmentation result, which means the algorithm is strongly confident on where it makes mistakes.

Other method like Valindria et al. (2017) used registration based method for quality assessment. It is a reliable method because it takes the prior of image by setting up a reference dataset. The problem of this method is inefficient testing. Every single case needs to do registration with all reference data to determine the quality but registration on 3D image is usually slow. Also the registration based method can hardly be transferred between datasets or modalities. Chabrier et al. (2006) and Gao et al. (2017) use unsupervised method to estimate the segmentation quality using geometrical and other features. However the application in medical settings is not clear. Also Robinson et al. (2018) tried a simple method using image-segmentation pair to directly regress the quality.

Kohlberger et al. (2012) introduced a feature space of shape and appearance to characterize a segmentation. The shape features in their system contain volume size, surface area, which are not necessarily related with the quality of the segmentation results. In our work we choose to learn a statistical prior of the segmentation mask and then determine the quality by how well a mask fits the prior. This is related with Out-of-Distribution (OOD) detection. Previous works in this field (Hendrycks & Gimpel, 2016) (Liang et al., 2017) made use of the softmax output in the last layer of a classifier to calculate the out-of-distribution level. In our case, however, for a segmentation method, we can only get a voxel-wise out-of-distribution level using these methods. How to calculate the out-of-distribution level for the whole mask becomes another problem. In addition, the segmentation algorithm can usually predict most of background voxels correctly with a high confidence, making the out-of-distribution level on those voxels less representative.

Auto-Encoder(AE), as a way of learning representation of data automatically, has been widely used in many areas such as anomaly detection (Zong et al., 2018), dimension reduction, etc. Variational autoencoder(VAE) (Kingma & Welling, 2013), compared with AE, can better learn the representation for the latent space. We employ VAE to learn the shape representation from the volumetric mask. Unlike method of Wu et al. (2015) which needs to pre-train with RBM, VAE can be trained following an end-to-end fashion. Qi et al. (2017) learned the shape representation from point cloud form, while we choose the volumetric form as a more natural way to corporate with segmentation task. Oktay et al. (2018) utilizes AE to evaluate difference between prediction and ground truth but not in an unsupervised way.

## 3 OUR EVALUATION METHOD

We first define our task formally. Denote the datasets we have as $(\mathcal{X}, \mathcal{Y})$, where $\mathcal{Y}$ is the label set of $\mathcal{X}$. We divide $(\mathcal{X}, \mathcal{Y})$ into training set $(\mathcal{X}_t, \mathcal{Y}_t)$ and validation set $(\mathcal{X}_v, \mathcal{Y}_v)$. Suppose we have a segmentation algorithms $F$ trained on $\mathcal{X}_t$. Usually we validate the performance of $F$ on $\mathcal{X}_v$ using $\mathcal{Y}_v$. Now we are doing this task without $\mathcal{Y}_v$. Formally, we try to find a function $L$ such that

$$\mathcal{L}(F(X), Y) = L(F, X; \omega) \tag{1}$$

where $\mathcal{L}$ is a function used to calculate the similarity of the segmentation result $F(X)$ respect to the ground truth $Y$, i.e., the quality of $F(X)$. How to design $L$ to take valuable information from $F$ and $X$, is the main question. Recall that the failure may happen when $X$ is a rare event. But to detect whether a image $X$ is within the distribution of training data is very hard because of the complex structure of image space, and actually that is what $F$ is trained to learn. In uncertainty based method Jungo et al. (2018) and Kwon et al. (2018), the properties of $F$ is encoded by sampling the

parameters of $F$ and calculating the uncertainty of output. The uncertainty does help predict the quality but the performance strongly relies on $F$. It requires $F$ to have Bayesian structure, which is not in our assumption and for a well-trained $F$. The uncertainty will mainly distribute on the boundary of segmentation prediction. So we change the formulation above to

$$\mathcal{L}(F(X), Y) = L(F(X); \omega) \tag{2}$$

By adding this constrain, we still take the information from $F$ and $\mathcal{X}$, but not in direct way. The most intuitive idea to do is directly applying a regression algorithm on the segmentation results to predict the quality. But the main problem is that the regression parameters trained with a certain segmentation algorithm $F$ highly relate with the distribution of $F(X)$, which varies from different $F$.

Following the idea of Kohlberger et al. (2012), we apply a two-step method, where the first one is to encode the segmentation result $F(X)$ into the feature space, and the second one is to learn from the feature space to predict the quality of $F(X)$. We propose a novel way of capturing the shape feature from $F(X)$, denoting as $S(F(X); \theta)$. Finally it changes to

$$\mathcal{L}(F(X), Y) = L(S(F(X); \theta); \omega) \tag{3}$$

## 3.1 Shape Feature of Variational Autoencoder

The shape feature is captured from Variational Autoencoder (VAE) trained with the ground masks $Y \in \mathcal{Y}_t$. Here we define the shape of the segmentation masks as the distribution of the masks in volumetric form. We assume the normal label $Y$ obeys a certain distribution $P(Y)$. For a predictive mask $\hat{y}$, its quality should be related with $P(Y = \hat{y})$. Our goal is to estimate the function $P(Y)$. Recall the theory of VAE, we hope to find an estimation function $Q(z)$ minimizing the difference between $Q(z)$ and $P(z|Y)$, where $z$ is the variable of the latent space we want encoding $Y$ into, i.e. optimizing

$$\mathcal{KL}[Q(z)||P(z|Y)] = E_{z \sim Q}[\log Q(z) - \log P(z|Y)] \tag{4}$$

KL is Kullback-Leibler divergence. By replacing $Q(z)$ with $Q(z|Y)$, finally it would be deduced to the core equation of VAE (DOERSCH, 2016).

$$\log P(Y) - \mathcal{KL}[Q(z|Y)||P(z|Y)] = E_{z \sim Q}[\log P(Y|z)] - \mathcal{KL}[Q(z|Y)||P(z)] \tag{5}$$

where $P(z)$ is the prior distribution we choose for $z$, usually Gaussian, and $Q(z|Y), P(Y|z)$ correspond to encoder and decoder respectively. Once $Y$ is given, $\log P(Y)$ is a constant. So by optimizing the RHS known as variational lower bound of $\log P(Y)$, we optimize for $\mathcal{KL}[Q(z)||P(z|Y)]$. Here however we are interested in $P(Y)$. By exchanging the second term in LHS with all terms in RHS in equation (5), we rewrite the training process as minimizing

$$\begin{aligned}
&E_{Y \sim \mathcal{Y}_t} \mathcal{KL}[Q(z|Y)||P(z|Y)] \\
&= E_{Y \sim \mathcal{Y}_t} \log P(Y) - E_{z \sim Q}[\log P(Y|z)] + \mathcal{KL}[Q(z|Y)||P(z)] \\
&= E_{Y \sim \mathcal{Y}_t} |\log P(Y) - S(Y; \theta)|
\end{aligned} \tag{6}$$

We denote $E_{z \sim Q}[\log P(Y|z)] - \mathcal{KL}[Q(z|Y)||P(z)]$ as $S(Y; \theta)$ for brevity. It shows that the training process is actually learning a function to best fit $\log P(Y)$ over the distribution of $Y$. After training VAE, $S(Y; \theta)$ becomes a natural approximation for $\log P(Y)$. So we just choose $S(Y; \theta)$ as our shape feature. In this method we use Dice Loss (Milletari et al., 2016) when training VAE, which is widely used in medical segmentation task. The final form of $S$ is

$$S(Y; \theta) = E_{z \sim \mathcal{N}(\mu(Y), \Sigma(Y))} \frac{2|g(z) \cdot Y|}{|Y|^2 + |g(z)|^2} - \lambda \mathcal{KL}[\mathcal{N}(\mu(Y), \Sigma(Y))||\mathcal{N}(0, 1)] \tag{7}$$

where encoder $\mu, \Sigma$ and decoder $g$ are controlled by $\theta$, and $\lambda$ is a coefficient to balance the two terms. The first term is the Dice's coefficient between $Y$ and $g(z)$, ranging from 0 to 1 and equal to 1 if $Y$ and $g(z)$ are equal. If we look at the shape feature closely, it indicates that after VAE is trained using data with only normal shape, the predictive mask $\hat{y}$ tends to be more likely in the distribution of normal shape if it can achieve less reconstruction error and is closer to prior distribution in the latent space, since $\log P(\hat{y}) \geq S(\hat{y}; \theta)$ holds all the time. On the other hand, for cases with high $P(\hat{y})$ but low $S(\hat{y}; \theta)$, it would introduce a large penalty to the object function (6), and is less likely to occur for a well-trained VAE.

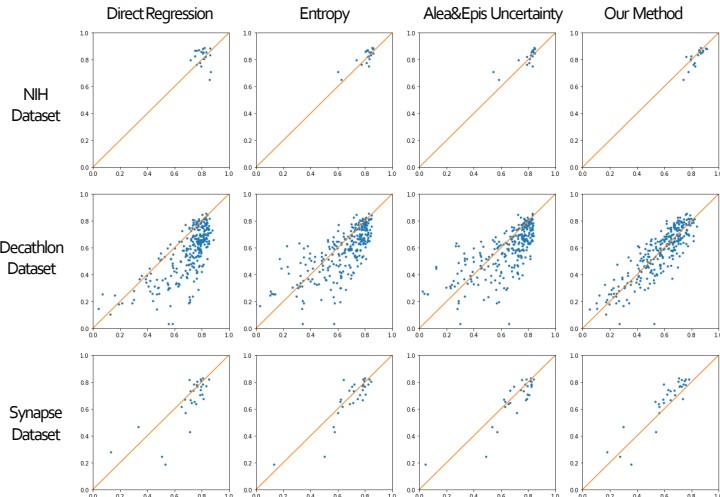

Figure 2: This figure shows our predictive Dice score (x axis) vs real Dice score (y axis). For each row, the segmentation algorithm is tested on the left most dataset. The four figures in each row show how the segmentation results are evaluated by 4 different methods.

## 3.2 SHAPE FEATURE FOR PREDICTING QUALITY

We assume that the shape feature is good enough to obtain reliable quality assessment. Intuitively thinking, for a segmentation result $F(X)$, the higher $\log P(F(X))$ is, the better shape $F(X)$ is in and thus the higher $\mathcal{L}(F(X), Y)$ is. Formally, taking the shape feature in section 3.1, we can predict the quality by fitting a function $L$ such that

$$\mathcal{L}(F(X), Y) = L(S(F(X); \theta); \omega) \tag{8}$$

Here the parameter $\theta$ is learned by training the VAE, using labels in the training data $\mathcal{Y}_t$, and is then fixed during the step two. We choose $L$ to be a simple linear model, so the energy function we want to optimize is

$$E(S(F(X); \theta); a, b) = ||aS(F(X); \theta) + b - \mathcal{L}(F(X), Y)||^2 \tag{9}$$

We only use linear regression model because the experiments show strong linear correlation between the shape features and the qualities of segmentation results. $\mathcal{L}$ is the Dice's coefficient, i.e. $\mathcal{L}(F(X), Y) = \frac{2|F(X) \cdot Y|^2}{|F(X)|^2 + |Y|^2}$.

## 3.3 TRAINING STRATEGY

In step one, the VAE is trained only using labels in training data. Then in step two $\theta$ is fixed. To learn $a, b$, the standard way is to optimize the energy function in 3.2 using the segmentation results on the training data, i.e.

$$\arg\min_{a,b} \sum_{(X,Y) \in (\mathcal{X}_t, \mathcal{Y}_t)} ||aS(F(X); \theta) + b - \mathcal{L}(F(X), Y)||^2. \tag{10}$$

Here the segmentation algorithm $F$ we use to learn $a, b$ is called the preparation algorithm. If $F$ is trained on $\mathcal{X}_t$, the quality of $F(X)$ would be always high, thus providing less information to regress $a, b$. To overcome this, we use jackknifing training strategy for $F$ on $\mathcal{X}_t$. We first divide $\mathcal{X}_t$ into $\mathcal{X}_t^1$ and $\mathcal{X}_t^2$. Then we train two versions of $F$ on $\mathcal{X}_t \setminus \mathcal{X}_t^1$ and $\mathcal{X}_t \setminus \mathcal{X}_t^2$ respectively, say $F_1$ and $F_2$. The optimizing function is then changed to

$$\arg\min_{a,b} \sum_{k=1,2} \sum_{(X,Y) \in (\mathcal{X}_t^k, \mathcal{Y}_t^k)} ||aS(F_k(X); \theta) + b - \mathcal{L}(F_k(X), Y)||^2. \tag{11}$$

In this way we solve the problem above by simulating the performance of $F$ on the testing set. The most accurate way is to do leave-one-out training for $F$, but the time consumption is not acceptable,

| | NIH Dataset | | | |
| --- | --- | --- | --- | --- |
| | MAE | STD | P.C. | S.C. |
| Direct Regression | 6.30 | 7.93 | -18.36 | -1.50 |
| Jungo et al. (2018) | 3.51 | 3.98 | 82.21 | 61.95 |
| Kwon et al. (2018) | 4.07 | 4.71 | **82.41** | 75.93 |
| Our method | **2.89** | **3.60** | 81.08 | **82.86** |
| | MSD Dataset | | | |
| | MAE | STD | P.C. | S.C. |
| Direct Regression | 14.47 | 12.50 | 72.26 | 70.17 |
| Jungo et al. (2018) | 11.86 | 16.31 | 71.24 | 77.71 |
| Kwon et al. (2018) | 12.68 | 18.31 | 70.42 | 77.77 |
| Our method | **7.00** | **9.14** | **86.23** | **85.02** |
| | MLC Dataset | | | |
| | MAE | STD | P.C. | S.C. |
| Direct Regression | 8.22 | 10.82 | 78.29 | 71.39 |
| Jungo et al. (2018) | 9.45 | 20.61 | 73.32 | 79.93 |
| Kwon et al. (2018) | 9.77 | 22.30 | 74.80 | 81.13 |
| Our method | **4.93** | **7.20** | **90.92** | **86.07** |

Table 1: Comparison between our method and baseline methods. The BNN is trained on NIH and tested on all three other datasets. Then, the segmentation results are evaluated by 4 methods automatically without using ground truth. Of the 4 methods, ours achieves the highest accuracy and the highest correlation between predicted Dice score and real Dice score.

and two-fold split is effective enough according to experiments. When the training is done, we can test on any segmentation algorithm $F$ and data $X$ to predict the quality $Q = aS(F(X); \theta) + b$.

## 4 EXPERIMENTAL RESULTS

In this section we test our alarm system on several recent algorithms for automatic pancreas segmentation that are trained on a public medical dataset. Our system obtains reliable predictions for the qualities of segmentation results. Furthermore the alarm system remains effective when the segmentation algorithms are tested on other datasets. We show better quality assessment capability and transferability compared with uncertainty-based methods and direct regression method. The quality assessment results are evaluated using mean of absolute error (MAE), stand deviation of residual error (STD), pearson correlation (P.C.) and spearman correlation (S.C.) between the real quality (Dice's coefficient) and predictive quality.

### 4.1 DATASET AND SEGMENTATION ALGORITHM

We adopt three public medical datasets and four recent segmentation algorithms in total. All datasets consist of 3D abdominal CT images in portal venous phase with pancreas region fully annotated. The CT scans have resolutions of $512 \times 512 \times h$ voxels with varying voxel sizes.

- **NIH Pancreas-CT Dataset (NIH).** The NIH Clinical Center performed 82 abdominal 3D CT scans(Roth et al., 2015) from 53 male and 27 female subjects. The subjects are selected by radiologist from patients without major abdominal pathologies or pancreatic cancer lesions.

- **Medical Segmentation Decathlon (MSD).** The medical decathlon challenge collects 420 (281 Training +139 Testing) abdominal 3D CT scans from Memorial Sloan Kettering Cancer Center. The subjects have cancer lesions within pancreas region[1].

---

[1]http://medicaldecathlon.com/index.html

| | 3D Coarse2Fine | | | | 3D V-Net | | | |
|-----|------|------|-------|-------|------|------|-------|-------|
| | MAE | STD | P.C. | S.C. | MAE | STD | P.C. | S.C. |
| NIH | 3.46 | 4.09 | 89.95 | 85.41 | 2.57 | 3.24 | 91.35 | 84.51 |
| MSD | 7.48 | 9.45 | 89.67 | 87.54 | 7.35 | 9.60 | 86.52 | 82.50 |
| MLC | 6.24 | 9.00 | 92.39 | 84.29 | 5.67 | 7.28 | 91.65 | 80.11 |
| | DeepLabV3 | | | | BNN | | | |
| | MAE | STD | P.C. | S.C. | MAE | STD | P.C. | S.C. |
| NIH | 5.35 | 5.83 | 63.34 | 78.80 | 2.89 | 3.60 | 81.08 | 82.86 |
| MSD | 9.18 | 10.80 | 85.79 | 81.87 | 7.00 | 9.14 | 86.23 | 85.02 |
| MLC | 6.23 | 7.06 | 94.84 | 89.63 | 4.93 | 7.20 | 90.92 | 86.07 |

Table 2: Different algorithms tested on different datasets are evaluated by our alarm system. Without tuning parameters, the system can be directly applied to evaluate other segmentation algorithms

- **Multi-atlas Labeling Challenge (MLC).** The multi-atlas labeling challenge provides 50 (30 Training +20 Testing) abdomen CT scans randomly selected from a combination of an ongoing colorectal cancer chemotherapy trial and a retrospective ventral hernia study [2].

The testing data for the last two datasets is not used in our experiment since we have no annotations for these cases. The segmentation algorithms we choose are V-Net (Milletari et al., 2016), 3D Coarse2Fine (Zhu et al., 2018), Deeplabv3 (Chen et al., 2018), and 3D Coarse2Fine with Bayesian structure (Kwon et al., 2018). The first two algorithms are based on 3D networks while the Deeplab is 2D-based. The 3D Coarse2Fine with Bayesian structure is employed to compare with uncertainty based method and we denote it as Bayesian neural network (BNN) afterwards.

## 4.2 BASELINE

We compare our method with three baseline methods. Two of them are based on uncertainty and the last one directly applies regression network on the prediction mask to regress quality in equation (2).

- **Entropy Uncertainty**. Jungo et al. (2018) calculated the pixel-vise predictive entropy using Bayesian inference. Then, the uncertainty is summed up over whole imagxe to get the doubt score and the doubt score would replace the shape feature in (8) to regress the quality. They sum is weighed by the distance to predicted boundary, which somehow alleviates the bias distribution of uncertainty. Their method is done in 2D image and here we just transfer it to 3D image without essential difficulty.

- **Aleatoric and Epistemic Uncertainty**. Kwon et al. (2018) divided the uncertainty into two terms called aleatoric uncertainty and epistemic uncertainty. We implement both terms and calculate the doubt score in the same way as Jungo et al. (2018) because the original paper does not provide a way. The two doubt scores are used in predicting the quality.

- **Direct Regression**. A regression neural network is employed to directly learn the quality of predictive mask. The training data for this network is the prediction of segmentation algorithm $F$ on $\mathcal{X}_t$ and the real Dice's coefficient between the predictive mask and label mask is used as the supervision.

## 4.3 IMPLEMENTATION DETAIL

For data pre-processing, since the voxel size varies from case to case, which would affect the shape of pancreas and prediction of segmentation, we first re-sample the voxel size of all CT scans and annotation mask to $1mm \times 1mm \times 1mm$. For training VAE, we apply simple alignment on the annotation mask. We employ a cube bounding box which is large enough to contain the whole pancreas region, centered at the pancreas centroid, then crop both volume and label mask out and resize it to a fixed size $128 \times 128 \times 128$. We only employ a simple alignment because the human pose is usually fixed when taking CT scan, e.g. stance, so that the organ will not rotate or deform heavily.

---

[2]https://www.synapse.org/!Synapse:syn3193805/wiki/217789

For a segmentation prediction, we also crop and resize the predictive foreground to $128 \times 128 \times 128$ and feed it into VAE to capture the shape feature.

During the training process, we employ rotation along x,y,z axes for $-10,0,10$ degree respectively and random translation for smaller than 5 voxel on annotation mask as data augmentation. This kind of mild disturbance can enhance the data distribution but keep the alignment property of our annotation mask. We tried different dimension of latent space and finally set it to 128. We found that with VAE with latent space of different dimension will have different capability in quality assessment. The hyper parameter $\lambda$ in object function of VAE is set to $2^{-5}$ to balance the small value of Dice Loss and large KL Divergence. We trained our network by SGD optimizer with batch size 4. The learning rate for training VAE is fixed to 0.1. We build our framework and other baseline model using TensorFlow. All the experiments are run on NVIDIA Tesla V100 GPU. The first training step is done in total 20000 iterations and takes about 5 hours.

## 4.4 PRIMARY RESULT AND DISCUSSION

We split NIH data into four folds and three of them are used for training segmentation algorithms and our pipeline; the remaining one, together with all training data from MSD and MLC forms the validation data to evaluate our evaluation method. First we learn the parameter of VAE using the training label of NIH dataset. Then we choose BNN as the preparation algorithm. The training strategy in section 3.3 is applied on it to learn the parameters of regression. For all the baseline methods, we employ the same training strategy of jackknifing as in our method and choose the BNN as preparation algorithm for fair comparison. Finally we predict the quality of predictive mask on the validation data for all the segmentation algorithms. Note that all segmentation algorithms are trained only on the NIH training set.

Table 1 reports the results of using three baseline models and our method to evaluate the BNN model tested on three datasets. In general, our method achieves the lowest error and variance on all datasets. In our experiment, the BNN achieves 82.15, 57.10 and 66.36 average Dice score tested on NIH, MSD and MLC datasets respectively. The segmentation algorithm trained on NIH will fail on some cases of other datasets and that is why we need the alarm system. The spearman coefficient for direct regression method on NIH dataset is close to 0 because the testing results on NIH are all of high quality and the regression result is not sensitive to slight variation in quality. Uncertainty based methods can better predict the quality but as shown in Figure 1, the uncertainty mainly distributes on the boundary of predictive mask but not on the missing parts or false positive parts. When the BNN is tested on the other two datasets, our method remains stable in predicting the quality. Table 2 shows the quality assessment results for 4 different segmentation algorithms. For each segmentation algorithm, When evaluating the segmentation results from DeepLab algorithm tested on MLC dataset, the accuracy is lower but the correlation between the predictive quality and real quality is high.

## 5 CONCLUSION

In the paper we present a VAE based alarm system for segmentation algorithms which predicts the qualities of the segmentation results without using ground truth. We claim that the shape feature is useful in predicting the qualities of the segmentation results. To capture the shape feature, we train a VAE using only ground truth masks. We utilize the fact that rare events will achieve larger value for loss function, and successfully detect the out-of-distribution shape according to the value for loss function in the testing time. In the second step we collect the segmentation results of the segmentation algorithm on the training data and extract the shape feature of them to learn the parameters of regression. By applying jackknifing training on the segmentation algorithm, we will get segmentation results of different qualities on the training data, therefore obtain more accurate regression parameters.

The reliable quality assessment results prove both that the shape feature capturing from VAE is meaningful and that the shape feature is useful for quality assessment in the segmentation task. Furthermore, our proposed method outperforms the uncertainty based methods and direct regression method, and possesses better transferability to other datasets and other segmentation algorithms.

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
