# OpenReview forum: "An Alarm System for Segmentation Algorithm Based on Shape Model"
_ICLR.cc/2019/Conference_

### Official Review · AnonReviewer3 · 2018-11-03
**Evaluates the quality of segmentation using  a shape prior learned from ground-truth masks with a variational autoencoder**

**Rating:** 5
**Confidence:** 4

**Review:**

This paper presents an approach to evaluate the quality of segmentations. To achieve this, a variational auto-encoder (VAE) is trained on the ground truth masks to extract shape-relevant information in the feature space, assuming that incorrect segmentations will be far from the normal distribution. Then, a regression model is trained to predict the quality of the segmentation based on the shape-learned features. The authors use several datasets focusing on pancreas segmentation to evaluate their quality-assessment approach, showing competitive performance with respect to other approaches.

The paper is well written, easy to follow in general, and the methodology is sound. Nevertheless, I have some concerns related to the applicability of this approach.

- Closely related works in the literature are missing:

There is a closely related recent work that used auto-encoders on the sets of ground-truth masks to build representations of shape and constrain the outputs of deep networks: Otkay et al., Anatomically Constrained Neural Networks (ACNN): Application to Cardiac Image Enhancement and Segmentation, IEEE TMI 2017

This work does not focus directly on quality assessment. However, I believe the loss in this work, which evaluates the difference between the obtained segmentation (characterized by the outputs of a deep network) and an auto-encoder description of shape, can be used directly as a criterion for evaluating the quality of segmentation (on validation data) in term of consistency with the shape prior. I think this work is very closely related and should be discussed.

Also, a quick google search provided some missing references related to this work. I think including comparisons to the recent work in [1], for example, would be appropriate. As the focus is on quality assessment of medical image segmentation, I would suggest a deeper review of the literature.

[1] Vanya V Valindria, Ioannis Lavdas, Wenjia Bai, Konstantinos Kamnitsas, Eric O Aboagye, Andrea G Rockall, Daniel Rueckert, and Ben Glocker. Reverse classification accuracy: Predict- ing segmentation performance in the absence of ground truth. IEEE Transactions on Medical Imaging, 2017.
[2] S. Chabrier, B. Emile, C. Rosenberger, and H. Laurent, “Unsupervised performance evaluation of image segmentation,” EURASIP Journal on Applied Signal Processing, vol. 2006, pp. 217–217, 2006.
[3] Gao H, Tang Y, Jing L, Li H, Ding H. A Novel Unsupervised Segmentation Quality Evaluation Method for Remote Sensing Images. Sensors. 2017 Oct 24;17(10):2427.

- The proposed quality assessment uses the learned shape features.  Even though it is strong prior information, there  might be situations where the predicted segmentation might be plausible in terms of shape, but is not a good segmentation.

-  I wonder how this approach works in problems with a high size/shape variation. For example, in the case of tumors, where their shape is unpredictable and each unknown case can be seen as a ‘rare’ example.

- To better capture the shape in the proposed approach, images need to be aligned, which limits the applicability of this approach to aligned volumes only.

- This approach gives a global hint about a given segmentation result, as a whole. I think it would be more interesting to provide local information on a segmentation, as it may happen that a predicted contour is generally correct, but there are some crispy borders in some points due to low contrast, for example. Even though the quality assessment would say that the prediction is correct, the contour may be unusable for certain applications, where a minimal surface distance is required (e.g., radiotherapy).

- As the quality assessment is based on shape and not in image information, it would be interesting to see how accurately it predicts the performance on different image modalities (for example, the method is trained on ground truth masks corresponding to CT images and quality is assessed in segmentations performed in MRI).

If I understood correctly, comparison with other methods is done with the same dataset under the same conditions (i.e., all the images are pre-aligned). As the other methods might not have the limitation of requiring aligned images, it would be interesting to compare also the performances in this situation.

How the training (or the VAE) is adapted for DeepLab-3, as it is based on 2D convolutions?

Minor: The paper needs a proof-read to fix some issues (e.g. ‘the properties of F is encoded’)

---

> ### Author Response · Authors · 2018-11-27
> **Response to Reviewer3**
>
> We thank the reviewer for the constructive feedback.
>
> Q1: Missing literatures.
> Thanks for sharing these related work with us.
> For the ACNN[1] work, they used an additional loss function to evaluate the difference between the encoded segmentation prediction and encoded label map to guide the training. It is obvious that this loss function can indicate the quality of a segmentation prediction, but to calculate this loss function requires a label map, while in our work we focus on evaluating the segmentation prediction WITHOUT ground truth. We don’t see the close connection between this work and ours.
> For the other literatures, we also add a paragraph in Section 2 to discuss the connection and difference. The main message is as follows: For [2], they use registration based method to do label propagation. It is a reliable method because it takes the prior of image by setting up a reference dataset. The problem of this method is inefficient testing. Every single case needs to be registrated with all reference data to determine the quality. However the registration on 3D image is usually very slow. In addition, the registration based method can hardly be transferred between datasets or modalities.
> For [3][4], they use unsupervised method to estimate the segmentation quality using geometrical and other features. However the application in medical settings is unclear, as is also mentioned in [2].
>
> Q2: Segmentation with good shape but low quality
> This situation indeed can appear theoretically. But it is very rare. In our experiment, only 1 out of 373 cases has relative plausible shape but very low dice score(See Figure 2, sub-figure line2 col4, one blue point is very far from the line y=x).
>
> Q3: High size/shape variation
> In this work, we focus on learning the prior of shape so we only do experiments on the organ which has stable shape. We show that this prior can be well learned and shared between different datasets, and can be used in predicting the segmentation quality without ground truth. Take tumor as an example, the prior of the texture may be more important so we may use other methods to deal with the texture prior in our future work.
>
> Q4: Image alignment
> The training data are first aligned.  However, during the training process, we alleviate the dependency on alignment by augmenting the data with rotation, translation and scaling etc.. During testing, no alignment is needed. As is shown in our experiments, our method achieves good performance and does not show a clear dependency on data alignment.
>
> Q5: local hint
> The prediction from the 3D segmentation methods is usually smooth so the local information here is not always useful. But for 2D segmentation methods like deeplab, they are trained using slices along the axial axis. So the whole 3D prediction may have unsmooth boundary along the axial axis. We think that is why the performance for evaluating deeplab is not as good as evaluating other segmentation algorithms(See Table2). It would be interesting to explore more in the future.
>
> Q6: Different modalities
> Our method is actually appearance independent. We show this by doing experiment cross different datasets. Although we only use datasets of CT scan, the difference between datasets is already big enough. The performance of a state-of-art segmentation algorithm trained on NIH will drop dramatically when tested on the other two datasets. But still our approach can give reliable quality prediction for the segmentation prediction on these two datasets. It shows the shape prior can be shared between datasets. So between modalities, as long as the shape domain doesn’t change, (i.e. between MRI and CT) the performance on MRI will still be the same.
>
> Q7: Adapted for DeepLab-3
> Although deeplab is a 2D based method, it can be applied to 3D CT scan slice by slice and generates a 3D segmentation mask finally. As our method only takes the segmentation mask as input, there is no problem adapting for deeplab.
>
> [1] Anatomically Constrained Neural Networks (ACNN): Application to Cardiac Image Enhancement and Segmentation
> [2] Reverse classification accuracy: Predicting segmentation performance in the absence of ground truth
> [3] Unsupervised performance evaluation of image segmentation
> [4] A Novel Unsupervised Segmentation Quality Evaluation Method for Remote Sensing Images

---

### Official Review · AnonReviewer1 · 2018-11-05
**Interesting and novel way of quantifying the quality of segmentations**

**Rating:** 6
**Confidence:** 4

**Review:**

This paper explores the idea of having a VAE modelling the probability distribution of the real segmentations, in order to quantify the quality of the predicted segmentation (using another network). The paper refines this idea by applying regression over two parameters. The overall idea is interesting and novel to the best of my knowledge. Experimental results look convincing.

The paper does a good job at presenting the motivation, reads well in general, and it is well written (except the paragraph Entropy Uncertainty in Sec. 4.2 which contains several typos).

Some comments:
S(F(X); θ) looks good enough as an estimator. It would be good to see how it does by itself, reporting that as an ablation experiment, assessing how important it is to carry out the second step (fitting a, b).
In the last paragraph of Sec. 2, I am not sure what it is meant by "Variational autoencoder(VAE) (Kingma & Welling, 2013), compared with AE, has stronger representation capability and can also serve as a generative model". No doubt about the latter point, but not sure about the former.
Sec. 3.3 is somewhat confusing, for example: what is E in eq. 9 should be L?

Revision: in light of the relevant papers brought up by AnonReviewer3 and AnonReviewer4, that have not been discussed in the paper, I modify my rating to 6.

---

> ### Author Response · Authors · 2018-11-27
> **Response to Reviewer1**
>
> We thank the reviewer for the constructive feedback.
>
> Q1: S(F(X); θ) itself as an estimator
> We have observed that S(F(X); θ) has strong linear correlation with the real quality but the value of them don’t exactly match, which is why we add a linear regression on S(F(X); θ).Only using S(F(X); θ) itself will change the MAE in Table1 of our method to 3.85, 7.62, 5.40 respectively.
>
>
> Q2: Confusing sentence in the paper
> Sorry for the misleading. By saying VAE, compared with AE, has stronger representation capability, we mean that VAE can better learn the statistical prior of the segmentation mask. For example, we have found that AE can better reconstruct the ground truth segmentation mask but will do worse than VAE in quality prediction, that is, AE tries to reconstruct everything perfectly, while VAE only tries to reconstruct the segmentation mask close to ground truth well. The difference here is because VAE adds a constraint in the latent space by using KL divergence while AE doesn’t have such constraint.
>
> Q3: Confusing notation
> Here L(S(F(X); θ);a,b)=a S(F(X); θ)+b. And E(S(F(X); θ);a,b)=||a S(F(X); θ)+b-L(F(X),Y)||2 represents for the loss function in the second step.
>
> Q4: Missing literatures.
> In our revised paper, we add some missing literatures that are related to our work and discuss the connection and difference.

---

### Official Review · AnonReviewer4 · 2018-11-16
**Interesting idea, but evaluation and relevant background literature is not thorough.**

**Rating:** 3
**Confidence:** 5

**Review:**

The authors present a method to detect poor quality segmentation results by using a VAE to understand the statistical distribution of segmentation masks, and detect outliers from that distribution in predictions. Method is compared to a few baselines to show improved results.

Pros:

1) The idea seems slightly novel, simple, and elegant, with respectable results.

Cons:

1) This method is related to Out-of-Distribution (OOD) detection, which is an entire field unto itself. None of the relevant literature around OOD has been covered by this paper, including several recent ICLR papers:

Hendrycks, Gimpel, "A BASELINE FOR DETECTING MISCLASSIFIED AND OUT-OF-DISTRIBUTION EXAMPLES IN NEURAL NETWORKS" ICLR 2017
Liang et al. "ENHANCING THE RELIABILITY OF OUT-OF-DISTRIBUTION IMAGE DETECTION IN NEURAL NETWORKS" ICLR 2018

2) The method is not compared to more naive approaches, such as building a network to take as input both modalities of original image and segmentation mask, and predict (classify) poor quality.

3) The method assumes segmentation masks have some strong statistical prior. This may be the case for organs, but can completely break down in other cases, such as skin lesion segmentation ( http://challenge2018.isic-archive.com ). In this circumstance, reviewer questions if more naive approach in (2) above would work better.


Reviewer believes authors have a good line of research, but that it requires additional literature review and experiments before it is ready for publication.


EDIT: Reviewer has considered the response by the authors. Key details of the baseline regressor are missing, such as the exact network structure used. As a result: 1) Reviewer is unable to determine if the baseline is a proper fair comparison. 2) Authors have confirmed the methods reliance on strong shape prior, but this caveat is not clearly mentioned in the paper as a requirement for the method to work. Furthermore, authors did not quantify what affect this reliance has by adding experiments on datasets with weak shape priors mentioned by reviewer. As a result, reviewer is lowering score. Reviewer encourages authors to continue this line of research, but carefully consider the feedback given to make the work stronger before publication.

---

> ### Author Response · Authors · 2018-11-27
> **Response to Reviewer4**
>
> We thank the reviewer for the constructive feedback.
>
> Q1: Missing literatures.
> Thanks for sharing these related work with us. In our revised paper, we add a paragraph in Section 2 to discuss the connection and difference. The main message is as follows: The existing methods you mentioned [1][2] made use of the softmax output in the last layer of a classifier to calculate the out-of-distribution level. In our case, however, for a segmentation method, we can only get a voxel-wise out-of-distribution level using your mentioned methods. Then how to calculate the out-of-distribution level for the whole case becomes another problem. Also, for most of background voxels, the segmentation algorithm will predict them as background very confidently, making the out-of-distribution level on those voxels less representative. The idea of using the activation value is similar with the uncertainty-based methods mentioned in our paper, so we expect them to have similar performance. Finally, using the softmax output from the classifier makes the out-of-distribution detector dependent on the classifier used, while our method can deal with different segmentation methods as shown in experiments.
>
>
> Q2: Compare to more naive approaches
> We have already tried a naïve approach (see Table 1, direct regression) which takes only segmentation mask as input and regress the quality indicator. That method doesn’t work very well. We also tried to take both original image and segmentation mask as input which  performs roughly the same . When conducting these two baseline methods, we firstly do rotation, translation and scaling on CT scan and test the segmentation algorithm on the augmented CT dataset to generate the predicted segmentation masks. Then we feed these masks into a regressor to predict the quality. When training the regressor, we have generated about 4000 3D segmentation masks as training data, but it still works not as well as our method.
>
> Q3:  For targets with larger shape variance
> In our work, we address a VAE-based method that can learn a statistical prior of the segmentation mask. We show  by experiment that this prior is important. It can guide the prediction of segmentation quality and can also be shared between different datasets, which is our main contribution.  For targets with large shape variance (e.g. lesion segmentation), how to effectively combine texture (or image) information is a promising research direction. However, with more texture information, it becomes harder to work crossing different datasets.
>
> [1] A BASELINE FOR DETECTING MISCLASSIFIED AND OUT-OF-DISTRIBUTION EXAMPLES IN NEURAL NETWORKS
> [2] ENHANCING THE RELIABILITY OF OUT-OF-DISTRIBUTION IMAGE DETECTION IN NEURAL NETWORKS

---

### Official Review · AnonReviewer2 · 2018-11-16
**Interesting approach and seems to have adequate convincing experiments**

**Rating:** 7
**Confidence:** 3

**Review:**

Summary:
      The paper tries to predict the quality of output of a segmentation algorithm applied to medical images. The approach of this paper is to looks at the "true" shape of the segmentation on the training samples and learn a VAE for the shape feature on them  for training samples.  For the test samples (that are new and are segmented only the algorithm whose quality is to be predicted), a linear function of the loss function of the learnt VAE applied to the output for the segmentation is used to predict quality. The linear function is tuned to the VAE loss of the output of the specific segmentation algorithm on the training samples.

 The basic premise is that VAE minimizes the gap between between the log likelihood of the true shape and the VAE loss function. Therefore, the gap should be small for "good" shapes while very bad for "bad/wrong" shapes. Therefore the VAE loss trained on the good shapes on the training examples can indicate the goodness of a segmentation algorithm's output.

Pros:
  I think the authors have compared to the number of baselines on three medical imaging datasets and show that their method via various metrics clearly outperforms others on this specific medical imaging application.

I like the primary technical idea behind the paper of detecting low quality outputs by projecting to the range space of a VAE and looking at its likelihood.

Cons:
1)  I dont know about the apriori assumption that shape of the segmentation will be the right feature to actually focus on. How general is this assumption for medical imaging tasks ?

2) Authors say - "Variational autoencoder(VAE) (Kingma & Welling, 2013), compared with AE, has stronger representation capability" - Why does the VAE have stronger representation capability?  - I dont understand this part.  Is it because it outputs the probabilities z given Y and Y given z that is somehow more useful ?

3) Can GANs be used instead of VAEs? Is there a natural loss function that could be used in this case during quality prediction?


Disclaimer: I am not an expert in the area for segmentation of medical images.

---

> ### Author Response · Authors · 2018-11-27
> **Response to Reviewer2**
>
> We thank the reviewer for the constructive feedback.
>
> Q1: Is the shape a right feature to focus on
> It is a useful feature for organ segmentation. We show this by experiment that it is more effective than other features.
>
> Q2: Confusing sentence in the paper
> Sorry for the misleading. By saying VAE, compared with AE, has stronger representation capability, we mean that VAE can better learn the statistical prior of the segmentation mask. For example, we have found that AE can better reconstruct the ground truth segmentation mask but will do worse than VAE in quality prediction, that is, AE tries to reconstruct everything perfectly, while VAE only tries to reconstruct the segmentation mask close to ground truth well. The difference here is because VAE adds a constraint in the latent space by using KL divergence while AE doesn’t have such constraint.
>
> Q3: Can GANs be used
> It would be interesting to try using GAN. We choose VAE because we think its latent space constraint may help which is proved to be true.

---

### Meta-Review · Area_Chair1 · 2018-12-11
**Promising direction of research for detecting poor quality segmentation, but further experiments and analysis must be completed.**

**Confidence:** 5
**Recommendation:** Reject

**Metareview:**

The authors present a method using a VAE to model segmentation masks directly. Errors in reconstruction of masks by the VAE indicate that the mask may be outside the distribution of common mask shapes, and are used to predict poor quality segmentation scenarios that fall outside the distribution of common segmentations.

Pros:
+ R2: Technical idea is interesting, and a number of baselines used to compare.
+ R1 & R4: Method is novel.

Cons:
- R3 & R4: The method ignores the original input in its prediction, making the method wholly reliant on shape priors. In situations where the shape prior is weak, the method may be expected to fail. Authors have confirmed this, but not added any experiments to quantify its effect.
- R4: The baseline regressor method is missing key details, which makes it impossible to judge if the comparison is fair (i.e. at minimum, number of learned parameters for each model, number of convolutional layers, structure of network, etc.). Authors have not provided these details. Authors have not investigated datasets with weak shape prior to see how methods compare in this setting.
- R2: GANs can be used as a baseline. Authors confirmed, but did not supply results.

Reviewers generally agree that the idea is novel, but the value of the approach cannot be determined due to missing baseline experiments, and missing details of baselines. Recommend reject in current form, but encourage authors to complete experiments.